



# Long-term behavior and stability of calibration models for NO and NO$_2$ low cost sensors

Horim Kim[1], Michael Müller[1,2], Stephan Henne[1], and Christoph Hüglin[1]

[1]Laboratory for Air Pollution and Environmental Technology, Empa, 8600 Dübendorf, Switzerland
[2]now at: Amt für Geoinformation, Kanton Basel-Landschaft, 4410 Liestal, Switzerland.

**Correspondence:** Christoph Hüglin (christoph.hueglin@empa.ch)

**Abstract.** Low-cost sensors are considered as exhibiting great potential to complement classical air quality measurements in existing monitoring networks. However, the use of low-cost sensors poses some challenges. In this study, the behaviour and performance of electrochemical sensors for NO and NO$_2$ were determined over a longer operating period in a real-world deployment. After careful calibration of the sensors, based on co-location with reference instruments at a rural traffic site during

six months and by using robust linear regression and random forest regression, the coefficient of determination of both types of sensors were high ($R^2 > 0.9$) and the root mean square error (RMSE) of NO and NO$_2$ sensors were about 6.8 ppb and 3.5 ppb, respectively, for 10 minute mean concentrations. The RMSE of the NO$_2$ sensors, however, more than doubled, when the sensors were deployed without re-calibration for a one year period at other site types (including urban background locations), where the range and the variability of air pollutant concentrations differed from the calibration site. This indicates a significant effect

of re-location of the sensors on the quality of their data. During deployment, we found that the NO$_2$ sensors are capable of distinguishing general pollution levels, but they proved unsuitable for accurate measurements, mainly due to significant biases. In order to investigate the long-term stability of the original calibration, the sensors were re-installed at the calibration site after deployment. Surprisingly, the coefficient of determination and the RMSE of the NO sensor remained almost unchanged after more than one year of operation. In contrast, the performance of the NO$_2$ sensors clearly deteriorated as indicated by a higher

RMSE (about 7.5 ppb, 10 minute mean concentrations) and a lower coefficient of determination ($R^2=0.59$).

## 1 Introduction

Severe negative impacts of urban air pollution on human health is still a major concern. Today, millions of city dwellers suffer from exposure to increased levels of air pollutants (Mage et al., 1996; Pascal et al., 2013; WHO, 2016). Nonetheless, existing air quality monitoring approaches are not always sufficient for a detailed understanding of urban air quality and human exposure

as large spatial and temporal variability of air pollutants challenges any monitoring (Marshall et al., 2008; Tan et al., 2014). Conventional instruments for air quality monitoring provide precise information on pollutant concentration and approach based on a limited number of point measurements. Their high acquisition and operational costs, as well as the requirement of specific expertise in operation of the instruments are, however, main obstacles for using them in larger numbers for achieving a denser spatial coverage of air pollutant observations in a city (Snyder et al., 2013; Kumar et al., 2015). Therefore, new techniques and



strategies for measuring urban air quality with higher spatial and temporal resolution are highly desirable. Low-cost sensors (LCS) have lately attracted a lot of attention as they have the potential to fill this gap. They are cost-effective, can be employed in large numbers, they are very simple to use and, in principle, require little maintenance (Karagulian et al., 2019).

LCS systems have already been used in various studies and their potential for air quality monitoring was demonstrated (e.g. Jiao et al., 2016; Cross et al., 2017; Hagan et al., 2018; Malings et al., 2019). However, the data quality that can be achieved with LCSs is often the main issue and a limiting factor. It is emphasized in multiple studies that high reliability of the sensors and appropriate calibration strategies are prerequisites for meaningful applications. Factors that can have a large influence on the data quality of LCSs for atmospheric measurements are the interference with ambient temperature and relative humidity (Bigi et al., 2018; Zimmerman et al., 2018), and insufficient sensitivity and selectivity. Low-cost sensors for reactive atmospheric gases have shown to be cross-sensitive to other gases, for example, electro-chemical sensors for nitrogen oxide (NO) were found to be cross-sensitive to nitrogen dioxide ($NO_2$) and ozone (Mead et al., 2013; Mueller et al., 2017).

Low-cost sensors need to be calibrated before they are used for atmospheric composition measurements (Castell et al., 2021). Laboratory calibration against reference material often has major drawbacks and calibration of LCSs is therefore predominantly done based on co-location with reference instruments operated in traditional air quality monitoring stations. Thus, the sensor output and relevant other environmental variables (e.g., temperature and relative humidity) are related to the true concentration values as represented by the reference measurement in parametric (e.g. Jiao et al., 2016; Kim et al., 2018; Malings et al., 2019) and non-parametric regression models (e.g. Cross et al., 2017; Hagan et al., 2018) or by using machine learning techniques (Bigi et al., 2018; Smith et al., 2019). The obtained (mathematical) relationship forms a calibration model that can be used for calibrating the sensors. Such a sensor calibration approach works generally well, although there are some challenging aspects that need to be taken into account. Firstly and for achieving a robust calibration model, it is important that during the co-located measurements all environmental conditions and the full concentration range of the measured pollutant, which the sensor will experience in a subsequent deployment, are covered (Castell et al., 2021). This requirement can often only be fulfilled through selection of a well suited reference station and a rather long duration of co-location measurements (Hagler et al., 2018), which may be in conflict with the rather short lifetime of air quality sensors. Secondly, it is currently unclear how long a sensor calibration model derived from co-location measurements can be applied and how often re-calibration of sensors needs to be done. Finally, the limited transferability of calibration models to new locations can be an important issue. This means that re-location of calibrated sensors may lead to data quality of the sensors that differs from that during co-location (Bigi et al., 2018).

In the present study, the above mentioned challenges related to the calibration of sensors by co-location measurements are investigated. Four low-cost sensor systems for measuring ambient NO and $NO_2$ were co-located to reference instruments for six months at a rural measurement location next to a motorway (Haerkingen site) with widely varying NO and $NO_2$ levels. After co-location, the LCSs were deployed for one year at four locations in the city of Zurich (Switzerland) co-located to $NO_2$ diffusion tube samplers for $NO_2$ sensor performance assessment. After deployment, the sensor units were brought back to the original co-location site (Haerkingen) where the LCSs were again measuring in side-by-side to reference instruments for another four months and for evaluation of the long-term stability of the employed calibration models.





## 2   Materials and Methods

### 2.1   Sensor unit

Four sensor units (denoted as AC009, AC010, AC011, AC012) utilized in this study were jointly developed by Empa and Decentlab GmbH. They were already utilized and described in detail in a previous study (Bigi et al., 2018). In each sensor unit, two identical electrochemical sensors for NO (Alphasense NO-B4) and $NO_2$ (Alphasense NO2-B43F), a relative humidity sensor and a temperature sensor (Sensirion STH21) are included (Bigi et al., 2018). The four electrochemical sensors in each sensor unit are denoted as NO_A, NO_B, NO2_A, and NO2_B in this study. All sensors recorded the measured data as 10-minute mean values, and the data were transmitted and saved with the corresponding timestamp in a database operated by Decentlab GmbH.

### 2.2   Co-location and deployment sites

The four sensor units were deployed next to reference instruments during two co-location campaigns. The first co-location campaign had a duration of 6 months (2018-06-29 - 2018-12-12; for AC009, the start date was 2018-07-25) and was done for sensor calibration and evaluation of sensor performance. The second 4 months long (2019-12-12 - 2020-03-31) co-location campaign was done about one and a half year after the first co-location campaign with the aim of assessing the long-term stability of calibrated sensors and re-evaluation of sensor performance after an extended time period of operation. During the time between the two co-location campaigns, the sensors have been deployed in a small sensor network in Zurich.

#### 2.2.1   Co-location site

The co-location measurements were done at the Haerkingen air-quality monitoring site. The Haerkingen site (HAE: 47.31° N, 7.82° E, 480 m.a.s.l) is part of the Swiss National Air Pollution Monitoring Network NABEL and situated 20 m north of a major highway (A1, 90'000 vehicles per day) in an open and rural environment. Thus, the concentrations of traffic related air pollutants like NO and $NO_2$ strongly depend on wind direction and traffic activity or daytime and span a wide concentration range (Bigi et al., 2018). Reference NO and $NO_2$ concentrations at HAE were measured as 10 minute mean values using a chemiluminescence instrument (T200, Teledyne Technologies Inc.), measurements of other air pollutants and meteorological variables are also available. As mentioned above, data from the first co-location campaign were used for finding the best calibration models and for evaluation of the performance of recently calibrated sensors. The data from the second campaign were used for evaluation of the long-term stability of the sensor calibration (i.e. the applicability of the calibration models determined during the first co-location campaign) and determination of changes of the performance of the sensors after deployment over an extended time period without re-calibration.





### 2.2.2 Deployment sites

In between the two co-location campaigns, the sensor units were deployed at four different locations in the city of Zurich for
measurement of the NO and NO$_2$ concentration during 11 months from 2018-12-13 to 2019-10-31. However, the data acqui-
sition of AC011 was paused from 2019-08-09, and that of AC010 from 2019-09-27 due to insects interrupting the air flow into
the sensor units. Geographical information including the site labels are presented in Table 1. An objective of this deployment
was to analyze the sensor performance in various locations in the city with different ranges of air pollutant concentrations.
It is highlighted that the sites ZSBS and ZMAN are urban traffic sites next to major roads in the city of Zurich. The total
traffic volume on the roads nearby ZSBS and ZMAN are 20,000 and 50,000 vehicles per day. Consequently, high NO and
NO$_2$ concentrations can be expected at these two sites. On the contrary, ZBLG is located in an urban green area surrounded
by residential buildings, and ZRIS is in a rural area on the outskirts of the city. Hence, it was expected that NO and NO$_2$
concentrations will be lower at this two sites compared with the two traffic sites.

NO$_2$ passive diffusion samplers (Palmes et al., 1976) were located at the four deployment sites close to the sensor units.
These four samplers are part of the NO$_2$ passive diffusion sampler network operated by the department of Environment and
Health Protection of the City of Zurich (UGZ). For comparison with the sensor data, integrated values of NO$_2$ concentrations
were available bi-weekly from 2018-12-04 to 2019-11-05. In total, 24 values of concentration pairs were considered at each site.
Even though the passive sampler produces data with insufficient temporal resolution (bi-weekly averaged), the observations
are known for their accuracy.

**Table 1.** Four deployment site of low-cost sensors. The coordinate system refers to the world geodetic system, WGS84, obtained from the
geo-mapping platform of the Swiss Confederation. © Swisstopo

| Street name (Abbreviation) | Site characteristic | Sensor | Deployment period | Coordinate system (WGS84) |
| --- | --- | --- | --- | --- |
| Im Ris (ZRIS) | Rural/Suburban | AC009 | 2018-12-13 - 2019-10-31 | 47° 19' 30.87" N 8° 30' 20.59" E |
| Bullingerhof 5 (ZBLG) | Residential | AC010 | 2018-12-14 - 2019-09-27 | 47° 22' 44.42" N 8° 30' 44.87" E |
| Seebahnstrasse 229 (ZSBS) | Urban-traffic | AC011 | 2018-12-13 - 2019-08-09 | 47° 22' 41.96" N 8° 31' 8.66" E |
| Manessestrasse 34 (ZMAN) | Urban-traffic | AC012 | 2018-12-13 - 2019-10-31 | 47° 22' 7.00" N 8° 31' 27.88" E |

### 2.3 Sensor calibration


Two calibration methods were utilized and evaluated in this study: Robust linear regression (Huber, 2004) and random forest
regression (Breiman, 2001). For finding a suitable calibration model for the individual NO and NO$_2$ sensors in a sensor unit,
the concentrations measured by the reference instrument have been used as target variables, the voltage of the corresponding
sensor ($V_{Sensor}$), the temperature ($V_{Tempertaure}$), and the relative humidity ($V_{Relative\ Humidity}$) as provided by the sensor unit have
been used as predictors. It is well known that low-cost sensors for measurement of atmospheric trace gases can be influenced
by external factors like temperature and relative humidity (Castell et al., 2021). In an earlier study by Mueller et al. (2017), it
was observed that the variation of relative humidity before the time of the measurement had an important effect on the data





quality of electrochemical sensors. Therefore, an additional variable, $D_{RH}$, was introduced for enhanced compensation of the effect of humidity on the sensor signal.

$$D_{RH} = \sum_{\Delta t=0}^{-500} \Delta S_{RH}(t + \Delta t) * exp(-\frac{\Delta t}{\Delta t_0}). \tag{1}$$

$\Delta S_{RH}$ represents the change in relative humidity, $\Delta t$ is the corresponding time lag in minutes and $\Delta t_0$ is a constant. Four different times for $\Delta t_0$ were chosen (60, 90, 120, and 150 minutes) for finding the best sensor calibration models.

### 2.3.1 Robust linear regression

Robust multiple linear regression is a commonly used method in the field of calibration. The effectiveness of the methodology
for calibration of air quality low-cost sensors has already been shown in several studies (e.g. Spinelle et al., 2015; Hagan et al., 2018). Robust regression is a technique that reduces the model distortion and bias induced by unusual observations and outliers (Andersen, 2008) by limiting their impact. The `rlm()` function from the R package `MASS` (Venables and Ripley, 2002, Ver. 7.3-54) was used for robust regression modelling.

### 2.3.2 Random forest regression

Random forest regression is an ensemble learning method in which numerous decision trees are examined to identify a superior model of a classification or regression. Each node in a decision tree is split by using the best option among a subset of predictors that are randomly chosen at that node (Breiman, 2001). Subsequently, multiple decision trees are ranked and the best option gets selected as an output. Random forest models showed great performance in previous studies, however, it was also shown that overfitting of the model may occur during calibration (Zimmerman et al., 2018). Moreover, the method can
not adequately predict values that are beyond the range of the training data set (Malings et al., 2019). In the present work, the `randomForest()` function in R package `randomForest` was employed for this modelling task. (Liaw and Wiener, 2002). For the random forest models the number of decision trees was set to 1000 and the chosen minimum number of observations in a terminal node was 100.

## 2.4 Evaluation of sensor performance

### 2.4.1 Model selection

Owing to the numerous possible combinations of predictor variables including their interactions in the above mentioned modelling approaches, a selection of 22 robust linear regression models and 9 random forest models were evaluated. The variables in each model are introduced in Table S1 and S2. Among the total of 31 calibration models, two models (1 robust linear regression model and 1 random forest model) were identified as the best performing models and selected for further investigation in
the study. For the evaluation, 80% of the sensor data from the first co-location campaign was randomly chosen and used for model training. The other 20% was applied for model testing. In every model, only a single validation with one training and





testing data was implemented because of the computational limitation. The selection was based on the normalized root mean square error (nRMSE) calculated for each model, and the models with the least nRMSE for both NO and $NO_2$ were chosen.

### 2.4.2 K-fold cross validation

Unlike the single validation process in the model selection, the chosen calibration models were then evaluated by K-fold cross-validation. The method is widely used to estimate the prediction error and the model accuracy. In the present study, the number of the fold (K) is 5 considering the literature recommendation (Rodriguez et al., 2010). For the validation, low-cost sensor data were randomly split into five different sub-groups. In each fold, four sub-groups of data (80% of total) were used as training data in each fold, and the remaining group was used for testing the model. As a result of the 5-fold cross-validation,

predictions of the five test data sets were obtained and combined into a single data set that was evaluated by comparison with the corresponding reference NO and $NO_2$ concentrations.

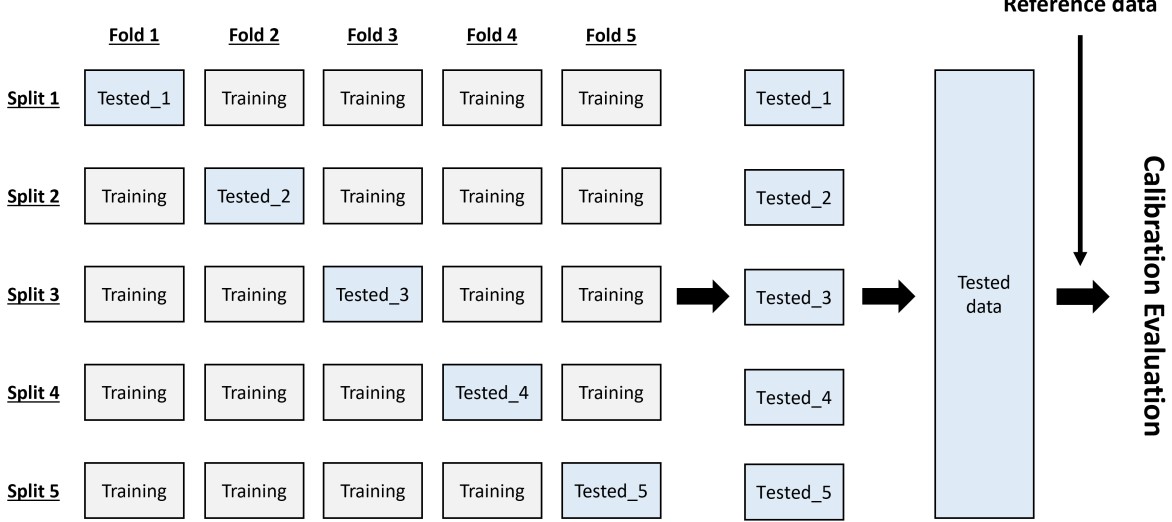

**Figure 1.** A scheme of the 5-fold cross validation utilized for the calibration model evaluation.

### 2.4.3 Evaluation approach

For evaluation of the sensor calibration performance, several statistical metrics (see table 2) in combination with target diagrams and taylor diagrams have been considered. The structure of target diagrams have been detailed in studies by Bigi et al. (2018)

and Zimmerman et al. (2018). The statistical measures were calculated using the `tdStats()` function in the *R* package `tdr` (Lamigueiro, 2018).





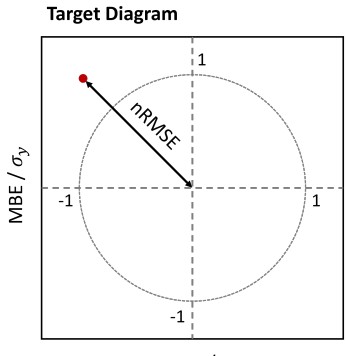

**Figure 2.** A schematic of the structure of the target diagram. The red point indicates the position of an exemplary point in the diagram.

**Table 2.** Statistical metrics visualized in target diagrams. In the terms in the third column, $\hat{y}_i$ represents a concentration measurement with a calibrated low-cost sensor and $y_i$ represents the corresponding concentration measurement with the reference instrument, $\sigma_{\hat{y}}$ and $\sigma_y$ are the empirical standard deviations of sensor and reference data.

| Metrics | Abbreviation | Term |
|---|---|---|
| Root mean square error | RMSE | $\sqrt{\dfrac{1}{n}\sum\limits_{i=1}^{n}(\hat{y}_i - y_i)^2}$ |
| Normalized root mean square error | nRMSE | $\dfrac{RMSE}{\sigma_y}$ |
| Mean bias error | MBE | $\dfrac{1}{n}\sum\limits_{i=1}^{n}(\hat{y}_i - y_i)$ |
| Centered root mean square error | CRMSE | $\sqrt{RMSE^2 - MBE^2}$ |

An schematic of a target diagram is presented in Figure 2 and its related statistical metrics are introduced in Table 2. The statistical metrics represented in target diagrams are the root mean square error (RMSE) and the mean bias error (MBE). The
RMSE is a non-negative measure for the error of the model predictions (or here the sensor measurements) and defined as the mean squared difference of the model predictions and the reference values. Similarly, the MBE is the average mean difference between predicted and reference values representing the mean bias of the model predictions. With the help of the MBE, the bias part in the RMSE can be corrected leading to a metric denoted as the centered root mean square error (CRMSE). For the target diagram, both CRMSE and MBE are normalized by the standard deviation of the reference values ($\sigma_y$) and plotted on
the x-axis and the y-axis, respectively. Note that for reference sites where the range of prevailing concentrations is high, $\sigma_y$ is also high and consequently normalised MBE and nRMSE tend at such sites to be smaller. Also note that CRMSE is generally positive, the sign of CRMSE/$\sigma_y$ in target diagrams is, however, determined by the sign of ($\sigma_{\hat{y}} - \sigma_y$). The interesting feature of target diagrams is that multiple information about the model behavior can easily be captured (Zimmerman et al., 2018). (i) A vector distance between the coordinate and the origin represents the normalized RMSE (nRMSE, RMSE/$\sigma_y$). (ii) (MBE/$\sigma_y$
> 0) and (MBE/$\sigma_y$ < 0) indicate model predictions that systematically overestimate or underestimate the reference. (iii) The standard deviation of the model prediction is larger (CRMSE/$\sigma_y$ > 0) or smaller (CRMSE/$\sigma_y$ < 0) than that of the reference data. (iv) The standard deviation of the model residuals is larger (outside of the circle of radius 1) or smaller (inside of the circle of radius 1) than that of the reference measurements.

In addition, Taylor diagrams are presented to visualize three statistical metrics: (1) Pearson's correlation coefficients ($r$) are
presented as the azimuthal angles from the y-axis. (2) Normalized standard deviations ($\sigma_{\hat{y}}/\sigma_y$) of models are given as the distance from the origin. (3) CRMSE (see Table 2) is proportional to the distance between the data points and the reference points on the x-axis (Taylor, 2001). The diagrams can demonstrate the metrics which target diagrams are not illustrated. Depending on the range of r, the shape of a diagram is either a semi-circle (-1≤ $r$ ≤1) or a quad (0≤ $r$ ≤1).



### 2.4.4 Sensor data filtering

The sensor systems had some obvious malfunctioning periods. The data acquired during such periods were eliminated and excluded from sensor performance evaluation. A malfunction period in this study has been defined as a period when the raw signal of an electrochemical sensor only recorded stable, non-fluctuating voltages and was only detected during the second co-location period. In contrast to the sensors, the corresponding measurements from the reference instrument during the identified malfunctioning periods showed temporally varying pollutant concentrations. Figure S15 illustrates an example of such a sensor
malfunctioning period. The reasons of sensor malfunctioning could, however, not be discovered. The exact time periods of malfunctioning sensors and a detailed description how erroneous sensor data has been identified and eliminated is given in the Supplementary Information (Section S-3.3.1).

## 3 Results and Discussion

### 3.1 Sensor calibration

The individual sensors for NO and $NO_2$ in the four sensor units were calibrated using the measurements of the first co-location campaign at the air quality monitoring site in Haerkingen. First, the calibration models with the most accurate prediction of NO and $NO_2$ were selected and the sensor performance was assessed. Second, the selected models were applied for the determination of the data quality of the sensors during the urban deployment and for re-assessment of the sensor performance during the second co-location campaign. In the present study, the air pollutant measurements are for the reason of practicality
and readability presented in parts-per-billion ($ppb$), although $nmol * mol^{-1}$ is a more accurate representation for the mole fractions of chemical species. For the same reason, the term concentration is also used when discussing mole fractions of measured air pollutants.

### 3.1.1 Selection of calibration models

A total of 22 robust linear regression (RLM) models and 9 random forest (RF) models (see Table S1 and S2) were evaluated and
the best RLM and RF models (in terms of normalised root mean square error) were selected for further investigation and sensor performance analysis. Figure 3 demonstrates that for calibration models, which include $D_{RH}$ as a predictor, the selection of $\Delta t_0$ = 150 minutes leads to the lowest mean nRMSE for both NO and $NO_2$. For the next step, models with $D_{RH}$ ($\Delta t_0$ = 150 minutes) and all other models without $D_{RH}$ as predictor were compared with respect to nRMSE. Figure 4 depicts the nRMSE of each model. It is prominent that the models that do not include the temperature signal (T) as predictor variable showed higher
nRMSE than the models including T. This is mainly due to the temperature dependence of the electrochemical sensors and the wide range of temperatures encountered during the first co-location period (-4°C to 35°C). In addition, including $D_{RH}$ in the model decreases the nRMSE of the $NO_2$ sensors more than including relative humidity (RH), while there is only a negligible difference for NO (see nRMSE between RLM_5 and 9, and between RF_6 and 8). The effect of relative humidity on the gas sensor was observed in previous research (Mueller et al., 2017). However, our current results emphasize that the temporal





variability or history of relative humidity is more influential than the present relative humidity itself. Overall, RLM_22 and
RF_6 models were chosen for further investigation in this study as they resulted in the smallest nRMSE in each class of models.
For $NO_2$ the two selected models clearly outperformed all other models, whereas for $NO$ other models showed similarly good
performances (Figure 4). The predictor variables for the two models are presented in Table 3.

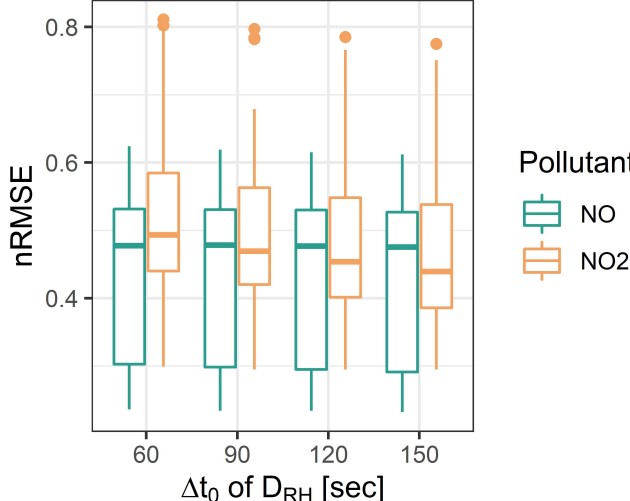

**Figure 3.** Boxplots of nRMSE by $\Delta t_0$ of $D_{RH}$. Five models (4 RLM and 1 RF model) were considered for a $\Delta t_0$, hence 40 data points are
contained in each boxplot.





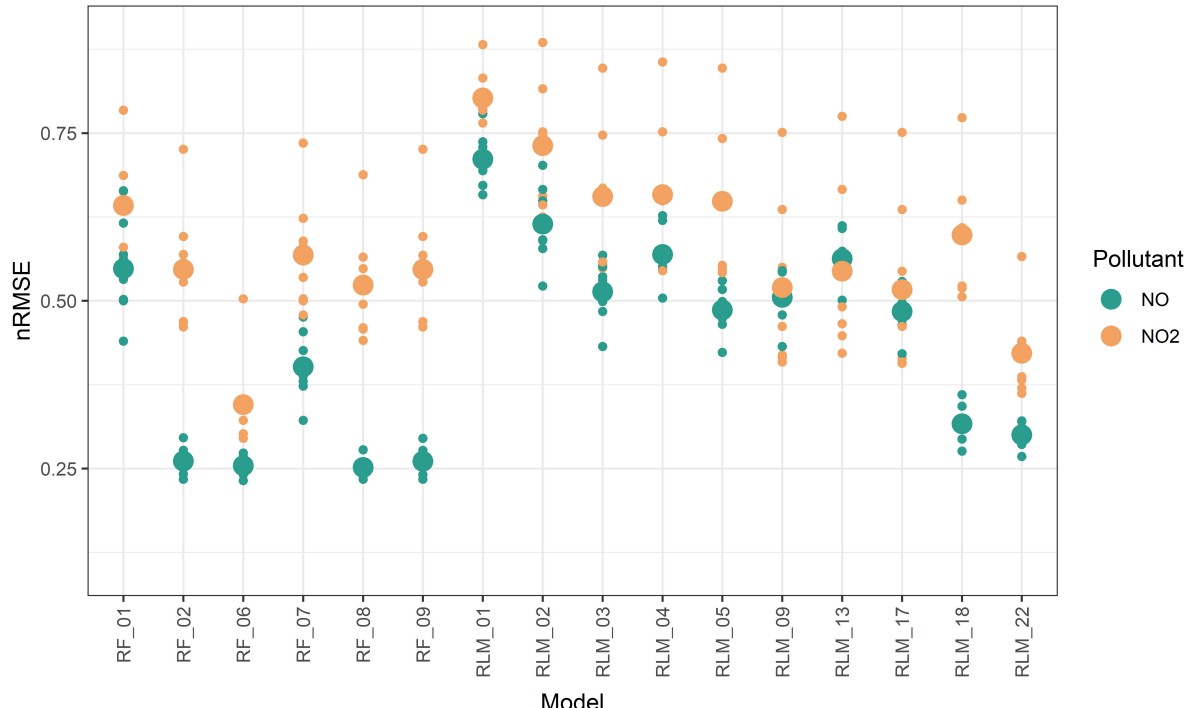

**Figure 4.** nRMSE values for each model. 8 data points from individual sensors of NO and NO$_2$ are shown separately and mean nRMSE values by pollutant are depicted in larger points.

**Table 3.** Model variables considered in the selected RLM and RF calibration model. Two models were denoted in the study as 'RLM' and 'RF' model.

| Model | Label | Model variables |
|---|---|---|
| RLM_22 | RLM | $V_{sensor}$, T, $V_{sensor}$*T, $D_{RH\_150}$ |
| RF_6 | RF | $V_{sensor}$, T, $D_{RH\_150}$ |

### 3.1.2 Calibration evaluation

The calibrated sensor data from the first co-location campaign were analysed and the different statistical metrics were calculated. Tables 4 and 5 provide the statistical metrics for sensor performance evaluation calculated as the mean of all sensors in the four sensor units and for concentration measurements with 10 minute time resolution. The considered statistical metrics and their graphical illustration in target and Taylor diagrams as shown in Figure 5 demonstrate that no significant difference between the performance of sensors calibrated using RLM or RF models can be found. The considered statistical metrics were

also calculated for different concentration ranges set at the rounded 25%, 50%, and 75% quantiles of the reference concentration measured at the Haerkingen site. The intention of this division into sub-groups was to investigate the sensor performance





for different air pollutant levels. The performance evaluation for such different sub-groups was motivated by the fact that for sensor calibration, a co-location site that covers the full range of the target pollutant concentration and of all other influential environmental variables is needed. Indeed, as shown in Figure 6(a), the sensors were exposed to a wide concentration range in Haerkingen station, from 0 ppb up to 232 ppb for NO and from 0.1 ppb up to 67 ppb for $NO_2$. However, for deployment in a sensor network, the sensors may be used at locations representing different site types and more narrow concentration ranges, including background environments.

The division demonstrated two important features that were not revealed when calculating the metrics from the full data set. First, at low concentrations (up to 25% quantile), nRMSE for the NO sensors was much higher (RLM: 11.23, RF: 8.35) than for the full concentration range (RLM: 0.30, RF: 0.26; see Table 4). The same behavior was observed for the $NO_2$ sensors, where nRMSE for the RLM was 1.80 (RF: 1.39) at low concentrations and 0.43 (RF: 0.30) at the full concentration range. This demonstrates a clear limitation of the LCSs utilized in this study; even though the sensors were frequently exposed to low target pollutant concentrations, the physical properties of the sensor (e.g. high sensor noise) fundamentally limits the measurement accuracy in this concentration range. Second, the mean bias error in each of the considered concentration ranges elaborate that the overall MBEs, which were close to zero, are actually the result of compensation between over-estimation at low NO and $NO_2$ concentrations and under-estimation at high concentrations. To be specific, the MBE values at low concentration (25% quantile) were 2-4 ppb, whereas those in the highest concentration quantile were similar but negative. The calibration models were able to amend the bias completely, and the residual plots in Figure S9 - S12 did not clearly reveal this behavior and did not clearly indicate any model deficiency. Instead, the opposite sign of the significant sensor bias at low and high concentration is masked when the overall MBE is considered, bearing the risk of misinterpretation of the sensor performance when deployed in a network and predominantly operating in a more narrow concentration range. Figure 5(a) and 5(c) also illustrate that MBE/$\sigma_y$ is near zero, but Figure 6(b) and 6(d) depict that MBE should be interpreted more carefully.

Furthermore, Figure 5(a) and 5(c) indicate that all the CRMSE/$\sigma_y$ values were negative in the first co-location period, which means that the standard deviation of the model prediction is smaller than that of the reference data. The low standard deviation in the predicted concentration is not surprising, because the prediction could not completely estimate the extreme value of concentration in the reference data. Same feature was identified in $\sigma_{\hat{y}}/\sigma_y$ of the highest concentration range (NO $\geq$ 28 ppb, $NO_2 \geq 26$ ppb), where the values are $\sim 1$ in both RLM and RF model (illustrated in Figure 6(c) and 6(e)). This implies that the predictions have small dynamics in this range, meaning that the extreme concentrations are poorly captured by the sensors. Moreover, the scatter plots in Figure S5 - S8 depict that RF models exhibit an upper limit in their predictions. The models cannot predict the concentration of NO above $\sim 130$ ppb, and that of $NO_2$ above $\sim 40$ ppb, and these limits lead to relatively lower standard deviations of RF model compared to RLM, as shown in the Figure 5(b) and 5(d). The main reason for this flaw is that RF models can only adequately predict within the concentration range that is covered by the training data. This deficiency was already reported previously (Bigi et al., 2018; Zimmerman et al., 2018).







**Figure 5.** Target diagrams (figure (a) and (c)) and Taylor diagrams (figure (b) and (d)) of each pollutant during the calibration evaluation period (1st) and the sensor re-location period after the deployment (2nd). In the target diagram, the centered root means square error (CRMSE) normalized by the standard deviation of reference data ($\sigma_y$) is stated on the x-axis, whereas the normalized mean bias error (MBE/$\sigma_y$) is stated on the y-axis. A distance from the origin indicates the nRMSE. A Taylor diagram illustrates three statistical metrics (Taylor, 2001): (1) Pearson's correlation coefficients ($r$) are presented as the azimuthal angles from the y-axis. (2) Normalized standard deviations ($\sigma_{\hat{y}}/\sigma_y$) of models are given as the distance from the origin. (3) CRMSE is proportional to the distance between the data points and the reference point (*Ref*) on the x-axis.



**Table 4.** Statistical metrics calculated for 10 minute concentration values from all NO sensors in the period of calibration evaluation (1st) and of the assessment after deployment (2nd). The sensor data were divided by rounded values of the 25%, 50%, and 75% quantile of the reference concentration of Haerkingen station in the first co-location campaign in order to keep a comparable number of data points in each sub-group. $R^2$ denotes the coefficient of determination.

| | RMSE [ppb] | | | | MBE [ppb] | | | | $R^2$ [-] | | | | nRMSE [-] | | | | $\sigma_{\hat{y}}/\sigma_y$ [-] | | | |
| | RLM | | RF | | RLM | | RF | | RLM | | RF | | RLM | | RF | | RLM | | RF | |
| Reference [ppb] | 1st | 2nd | 1st | 2nd | 1st | 2nd | 1st | 2nd | 1st | 2nd | 1st | 2nd | 1st | 2nd | 1st | 2nd | 1st | 2nd | 1st | 2nd |
|---|---|---|---|---|---|---|---|---|---|---|---|---|---|---|---|---|---|---|---|---|
| X < 2 | 6.50 | 4.89 | 4.83 | 5.38 | 2.20 | -2.24 | 4.00 | 4.42 | 0.02 | 0.14 | 0.01 | 0.00 | 11.23 | 8.90 | 8.35 | 9.79 | 10.64 | 8.22 | 4.66 | 5.47 |
| 2 ≤ X < 10 | 6.92 | 4.78 | 3.82 | 4.58 | 2.46 | -0.71 | 1.86 | 2.17 | 0.05 | 0.30 | 0.16 | 0.17 | 2.93 | 2.05 | 1.61 | 1.97 | 2.77 | 2.39 | 1.47 | 1.88 |
| 10 ≤ X < 28 | 7.76 | 6.40 | 6.39 | 5.83 | -1.60 | 0.44 | -2.17 | 0.28 | 0.24 | 0.41 | 0.34 | 0.42 | 1.52 | 1.25 | 1.25 | 1.14 | 1.69 | 1.63 | 1.43 | 1.49 |
| X ≥ 28 | 10.12 | 11.78 | 10.44 | 7.85 | -4.08 | 6.20 | -4.13 | -0.23 | 0.90 | 0.93 | 0.88 | 0.92 | 0.36 | 0.44 | 0.37 | 0.29 | 1.05 | 1.22 | 0.96 | 0.97 |
| Entire range | 7.93 | 7.63 | 6.84 | 6.08 | -0.23 | 1.05 | -0.03 | 1.52 | 0.91 | 0.96 | 0.94 | 0.95 | 0.30 | 0.29 | 0.26 | 0.23 | 0.96 | 1.18 | 0.91 | 0.96 |

**Table 5.** Statistical metrics calculated for 10 minute concentration values from all $NO_2$ sensors in the period of calibration evaluation (1st) and of the assessment after deployment (2nd). The sensor data were divided by rounded values of the 25%, 50%, and 75% quantile of the reference concentration of Haerkingen station in the first co-location campaign in order to keep a comparable number of data points in each sub-group. $R^2$ denotes the coefficient of determination.

| | RMSE [ppb] | | | | MBE [ppb] | | | | $R^2$ [-] | | | | nRMSE [-] | | | | $\sigma_{\hat{y}}/\sigma_y$ [-] | | | |
| | RLM | | RF | | RLM | | RF | | RLM | | RF | | RLM | | RF | | RLM | | RF | |
| Reference [ppb] | 1st | 2nd | 1st | 2nd | 1st | 2nd | 1st | 2nd | 1st | 2nd | 1st | 2nd | 1st | 2nd | 1st | 2nd | 1st | 2nd | 1st | 2nd |
|---|---|---|---|---|---|---|---|---|---|---|---|---|---|---|---|---|---|---|---|---|
| X < 9 | 3.77 | 10.49 | 2.91 | 10.87 | 1.97 | 8.46 | 1.62 | 9.39 | 0.17 | 0.00 | 0.26 | 0.00 | 1.80 | 5.34 | 1.39 | 5.53 | 1.65 | 3.03 | 1.29 | 2.62 |
| 9 ≤ X < 17 | 3.31 | 7.11 | 2.71 | 6.61 | 0.10 | 3.77 | 0.29 | 4.50 | 0.26 | 0.03 | 0.37 | 0.06 | 1.45 | 3.12 | 1.19 | 2.90 | 1.68 | 2.63 | 1.49 | 2.14 |
| 17 ≤ X < 26 | 4.00 | 6.27 | 3.27 | 4.69 | -0.60 | 0.75 | -0.23 | 0.89 | 0.30 | 0.10 | 0.37 | 0.12 | 1.54 | 2.44 | 1.26 | 1.83 | 1.81 | 2.55 | 1.58 | 1.88 |
| X ≥ 26 | 7.38 | 6.92 | 4.63 | 5.92 | -2.89 | -0.82 | -1.88 | -3.15 | 0.34 | 0.37 | 0.63 | 0.39 | 1.10 | 1.17 | 0.69 | 1.00 | 1.18 | 1.45 | 0.97 | 0.95 |
| Entire range | 4.86 | 7.91 | 3.45 | 7.45 | -0.31 | 3.25 | -0.02 | 3.27 | 0.82 | 0.55 | 0.91 | 0.59 | 0.43 | 0.76 | 0.30 | 0.72 | 0.91 | 0.91 | 0.92 | 0.72 |





**(a)** Distribution of reference NO and NO2 concentration in the first colocation period

**(b)** Target diagram : NO

**(c)** Taylor diagram: NO

**(d)** Target diagram : NO2

**(e)** Taylor diagram: NO2

**Figure 6.** Statistical metrics calculated for each reference concentration sub-group from the first co-location period. The sub-groups were defined by rounded values of 25%, 50%, and 75% quantile of the NO and $NO_2$ concentration during the period. (a) represents the density plots of pollutants concentration; (b) and (d) elaborate the target diagrams of each concentration range in both co-location periods; (c) and (e) illustrated the Taylor diagrams for same concentration ranges.



## 3.2 Sensor performance

### 3.2.1 Deployment period

After the first co-location period, the four sensor units were relocated and deployed at four sites in Zurich. Figure 7 illustrates the comparison between the measurements with the $NO_2$ sensors and the $NO_2$ passive samplers at the four deployment sites. It should be mentioned again, that two of the deployment sites, ZRIS and ZBLG, were located in suburban and urban background locations as shown in Figure S4(a) and S4(c), with relatively low concentrations of $NO_2$ compared to the two other locations

(ZSBS and ZMAN), which were strongly influenced by emissions from nearby road traffic. At ZRIS, where sensor unit AC009 was deployed, the bi-weekly $NO_2$ concentration measured with the passive samplers were 2-9 ppb, and at ZBLG (AC010) $NO_2$ concentration ranged between 6-22 ppb. At the two urban traffic locations ZSBS (AC011) and ZMAN (AC012), the $NO_2$ concentrations were higher and ranged from 19-31 ppb (ZSBS) and 33-48 ppb (ZMAN), respectively. In general, AC011 agreed well with the passive sampler data for both calibration models, although the sensor measurements were biased high

by on average 3.5 ppb (RLM) and 2.4 ppb (RF) (Figure 7 and Figure 8). For the suburban and the urban background sites ZRIS and ZBLG, the measurements with the sensor units (AC009 and AC010) showed clear mean biases for both calibration modelling approaches. Finally, at the highest polluted site ZMAN, the RF models resulted in substantial under-estimation of the passive sampler data (Figure 7), also visible in the time-series in Figure S13. The observed under-estimation of peak $NO_2$ concentrations by RF models results from the fact that predictions cannot go beyond the concentration range that has been

covered by the training data. Compared to RF models, the RLM calibration applied to AC012 resulted in a much smaller bias, although the scattering between sensor and passive sampler data was high as expressed by a large CRMSE (Figure 8). Figure 7 shows that the used LCSs for $NO_2$ enable a distinction between lower (ZRIS and ZBLG) and higher polluted sites (ZSBS and ZMAN) and therefore a general differentiation of locations with regard to the prevailing $NO_2$ levels. However, the limitations of the used LCSs for providing accurate measurements of the bi-weekly $NO_2$ are clearly visible. This can also be seen from Figure

8, where RMSE, CRMSE and MBE are shown for the co-location and the deployment periods. For comparison, the statistical metrics provided in Figure 8 were for the two co-location periods also calculated from bi-weekly averaged data. Figure 8(a) illustrates that the RMSEs during the deployment period were much larger than during the first co-location period, which was mainly resulting from an increasing bias (MBE) as discussed above, but also from increasing CRMSE. This observation implies an effect of re-location on the performance of the sensors, i.e. a change in the data quality provided by the sensors

when they are operated in other locations than the location chosen for co-location measurements and sensor calibration. A reason for the observed decreasing data quality during the deployment was likely the smaller range of $NO_2$ concentrations prevailing at the deployment sites compared to the co-location site, in particular for the suburban and urban background sites ZRIS und ZBLG. As also seen in the co-location period, the applied sensor calibration models tended to over-estimate $NO_2$ at low concentrations (Tables 4 and 5). Another reason for the decreasing data quality of the sensors during the deployment might

be differing combinations of influencing external factors like air temperature, relative humidity and interfering gases during deployment, leading to a reduced applicability of the calibration models and possibly larger measurement errors as expressed




by the higher CRMSE. Figure 8 also implies that for achieving the best possible data quality, implementation of strategies for the detection and the correction of sensor bias during deployment are needed.

290  Finally, we find that in agreement with the first co-location period, the RF calibration model generally performed slightly better than the RLM model. This can be seen in the somewhat smaller RMSE of the sensors calibrated using RF models and also the lower CRMSE in particular for the sensors that were deployed at the higher polluted sites (AC011 and AC012). However, AC012 showed a highly negative MBE, indicating the earlier mentioned inability for correctly predicting air pollutant concentrations for conditions, that have not been covered in the data used for training the calibration model. This should be kept in mind when sensors are calibrated using this modelling approach.

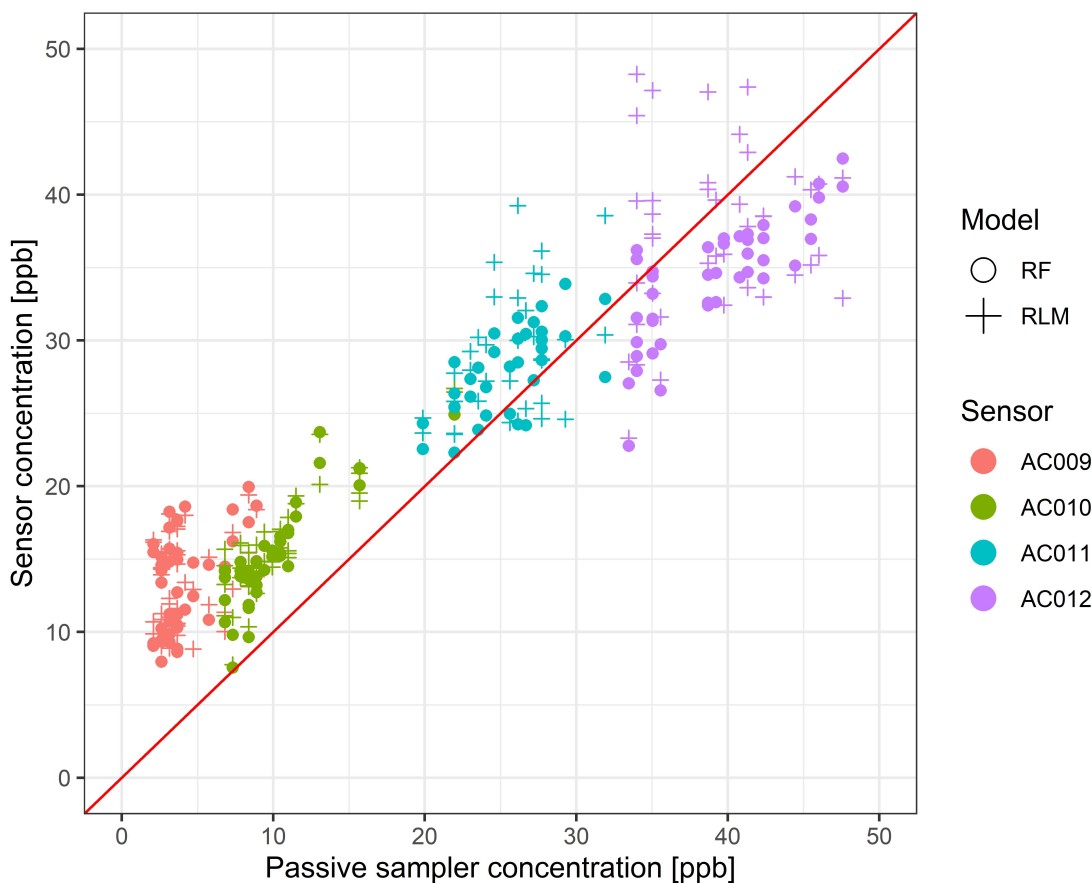

**Figure 7.** $NO_2$ sensor performance during the deployment in Zurich illustrated by the sensor concentration comparison to that of $NO_2$ passive samplers in each sites.





**Figure 8.** RMSE and CRMSE (a) as well as MBE (b) for $NO_2$ during the co-location periods and the deployment period. The metrics for the co-location periods are also calculated from from bi-weekly averaged data in order to facilitate the comparison with sensor performance during deployment.

### 3.2.2 Assessment of sensor performance after deployment

After the deployment at the four locations of the small sensor network in Zurich, the sensors were again installed at the co-location site Haerkingen for 4 months (2019-12-12 - 2020-03-31). During the second co-location period we observed some distinct sensor malfunctioning for several short periods. Hence, data filtering was implemented prior to performance analysis. Figure 9 illustrates that the data obtained during the identified malfunctioning periods caused severe under-estimation of the true NO and $NO_2$ concentrations and therefore had to be removed. Specific malfunctioning periods for each sensor unit are





elaborated in Table S6. An exact cause or single influence factor for sensor malfunctioning could not be identified. However, it is hypothesised that specific weather conditions may be the reason for the sensors not working properly. Meteorological data collected at the co-location site indicates that rain events occurred before and after most of the time periods that had to be filtered. Electrochemical sensors are known for their vulnerability towards humidity. Thus, the penetration of raindrops into the

sensor units may cause significant disturbance of the sensor signal. Nevertheless, rain events were not the sole factor for the sensor malfunction, because rain also occurred during the first co-location campaign without noticeable sensor malfunctioning, and rain events during the second co-location campaign did not always result in erroneous sensor signals. In addition, the possibility of interference by low battery was checked, however, none of the issues were detected from sensor log files. It is therefore speculated that an interaction between rain events and other meteorological factors such as wind speed caused the

sensor malfunctioning. However, because the exact reasons for the erroneous sensor data remain unknown, the applied data filtering was therefore based on visual screening of the sensor signal.

The air pollutant concentrations reported by the sensors were for the second co-location period calculated using the models developed in the first co-location campaign and applied during deployment. Tables 4 and 5 present the statistical metrics for hourly concentration measurements and a comparison with the values from the first co-location period. Surprisingly, the sensors

showed for NO after more than a year of operation still a comparable performance. The average RMSE of the 10 min NO sensor measurements slightly decreased from 7.9 ppb to 7.6 ppb for the RLM models and from 6.8 ppb to 6.1 ppb for the RF models. The target and Taylor diagrams in Figure 5 depict no substantial change in the shown metrics calculated from the NO measurements. In contrast to NO, the performance of the $NO_2$ sensors clearly decreased over this extended time period of operation, for example the average RMSE for 10 minute $NO_2$ measurements increased by 3-4 ppb for both calibration models

(Table 5 and Figure 8). The target diagram for $NO_2$ in Figure 5 indicates that this increase in RMSE is due to both, an increase in mean bias and an increase in the random component of the error as expressed by the CRMSE. The latter is also visible in the Taylor diagram for $NO_2$ and the reduced correlation coefficient during the second co-location period. In addition, the target and Taylor diagrams separated for the different quartiles of observed 10 minute mean values (Figure 6) show that the above described change in sensor performance over the extended time period is also visible when the different concentration ranges

are separately evaluated.





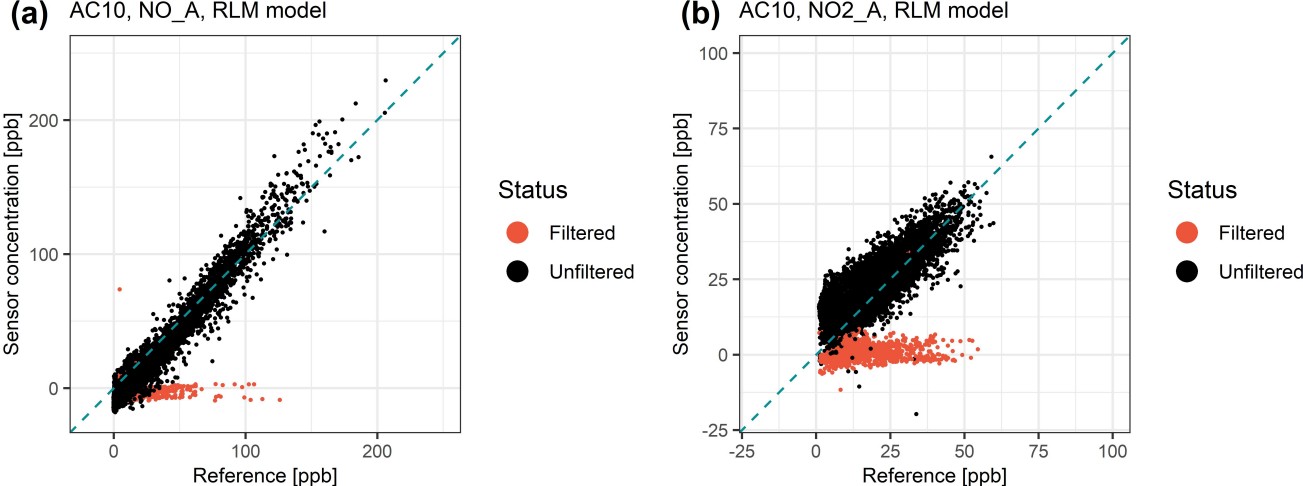

**Figure 9.** Scatter plots of sensor concentrations calibrated using the RLM model versus reference measurements during the second co-location period for (a) NO and (b) NO2 from AC010 (10 minute values). Sensor malfunctioning periods were visually identified and filtered (red dots). Similar scatter plots for the other sensor units can be sound in the supplementary material (Figure S16 - S19).

# 4   Conclusion

In this study, some of the main difficulties associated with the use of low-cost sensors for measuring air quality were investigated. In particular we analysed calibration, long-term stability of the sensor output, and the effect of re-location on sensor performance, i.e. the change of sensor behavior when used at a different location than the site used for calibration through
co-location with a reference instrument. Although only two specific types of sensors were used in this study, some general conclusions can be drawn. Co-location with reference instruments is a pragmatic and appropriate approach for the calibration of individual low-cost sensors. However, the duration of the co-location measurements should be sufficiently long so that a wide range of environmental conditions, which may occur during deployment, are covered. In addition, the chosen co-location site should allow to cover the full concentration range expected during deployment. Otherwise, the calibration model extrapolates
to conditions that have not been covered in the data used for training, leading to higher uncertainty and for some approaches (e.g., random forest regression) to significant bias.

In this study, duration and site of co-location were been chosen accordingly. The sensors were calibrated using two widely-used statistical approaches and the corresponding sensor performance was evaluated. During co-location with reference instruments, the sensors showed no overall bias and had a rather small CRMSE, when the full data set was analysed. However, when
the performance metrics were calculated for different concentration ranges (i.e., the quartiles of the observed concentrations), it was observed that the applied calibration models led to sensor measurements that were biased high at low concentrations and biased low at high concentrations. A similar behavior was observed for the $NO_2$ sensors when deployed in a small sensor network in the city of Zurich. In this case, the data quality of the sensors was much lower than expected from their perfor-



mance during co-location with a reference instrument. For a relatively clean city like Zurich, the achieved data quality was

not sufficient for meaningful quantitative measurements of $NO_2$. However, the sensors were capable of distinguishing between locations with lower, medium and higher $NO_2$ levels. An important factor for lower than expected data quality was seen in the fact that sensors were typically deployed in locations where the concentration range of the target air pollutant was considerably smaller than at the co-location site (e.g., at urban background locations). The calibration models derived from co-location with reference instruments might strongly be influenced by measurements at the highest prevailing concentrations and might

therefore not be optimal for cleaner locations.

Another important limitation of low cost air quality sensors may be their lifetime and the frequency of re-calibration. For the electrochemical sensors for NO we found no change in response behavior over a time period of more than 18 months and the data quality was therefore constant over time. In contrast, the electrochemical sensor for $NO_2$ showed decreasing performance over time and frequent interventions such as re-calibration or replacement may be needed for achieving the best possible data

quality. After about 18 month of deployment, the electrochemical sensors started to malfunction sporadically and during shorter time periods. Although the exact reasons remained unknown, this behavior might indicate ageing effects of the sensors itself or of other parts of the sensor unit. The occurrence of these malfunctions with increasing time of use might indicate that the quality control of sensors deployed in networks need to be strengthened over time.

*Code and data availability.* The observations data (sensors and reference), the calibration models, and all data analysis codes (both based on

R programming language) are available upon request from the authors. The data and codes will be made available on a public data repository at the time of the final revision of the manuscript.

*Author contributions.* HK and MM developed the raw data acquisition from the Decentlab GmbH and the sensor calibration models. MM established the field measurement of the sensors by organizing the sensor co-location measurements at the Haerkingen site and deployed the sensors in Zurich. HK implemented the pre-processing of the raw data, sensor calibration, post-processing of the prediction and carried out

the validation and data analysis for the study. HK performed the results and wrote the manuscript. CH and SH supervised the whole study, actively involved in writing the manuscript. All authors reviewed and agreed to the published version of the paper.

*Competing interests.* The authors declare that they have no conflict of interest.

*Acknowledgements.* The support and the data from the Swiss national air pollution monitoring network NABEL (FOEN/Empa) is gratefully acknowledged.





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
