# Peer review of "Long-term behavior and stability of calibration models for NO and NO2 low cost sensors"

_Atmospheric Measurement Techniques, 2021_

## Author Comment (AC1)

**Reply to the comments of Anonymous Referee #1 on the manuscript "Long-term behavior and stability of calibration models for NO and NO$_2$ low cost sensors"**

Horim Kim[1], Michael Müller[1,2], Stephan Henne[1], and Christoph Hüglin[1]

[1]Laboratory for Air Pollution and Environmental Technology, Empa, 8600 Dübendorf, Switzerland
[2]now at: Amt für Geoinformation, Kanton Basel-Landschaft, 4410 Liestal, Switzerland.

We sincerely thank the reviewer for the positive review and the constructive feedback to improve the manuscript. Please find the detailed replies to all points made by the reviewer. In the following, the reviewer's comments are given in black and the replies in blue and the revised text in green.
* * *
This manuscript evaluated NO and NO$_2$ sensors' performance during long-term deployments. Calibration models and evaluation metrics are described in detail, supporting relative conclusions. The manuscript is organized well, and this topic is important for the field deployment of air quality sensors. Therefore, I would recommend accepting the manuscript after minor revision.

1. It is good to use the Taylor diagram to show multiple metrics. It will be helpful to describe where an ideal sensor should locate in the Taylor diagram.

**Reply:** An ideal model/sensor would have Pearson's correlation coefficients ($r$) = 1 and normalised standard deviation $\sigma_{\hat{y}}/\sigma_y$ = 1. It would be located at the point marked as Ref in Figure 5. We already described the location of this reference in the caption, but reworded to clarify the meaning of the reference point.
The reference point gives the location of a perfect model/sensor.

2. On Page 5, please explain more about equation 1. It is unclear why the author would like to address relative humidity in this form. In addition, more information is needed regarding the importance of $\Delta t_0$.

**Reply:** First and foremost, there was a typo in the equation. Previously the exponential term was $exp(-\frac{\Delta t}{\Delta t_0})$, but the exponent should have been without minus (-). We now corrected the equation in the revised paper as below:

$$D_{RH} = \sum_{\Delta t=0}^{-500} \Delta S_{RH}(t+\Delta t) * exp(\frac{\Delta t}{\Delta t_0}).$$

(1)

As mentioned in Section 2.3, Equation (1) was investigated in the earlier study (Mueller et al.: Design of an ozone and nitrogen dioxide sensor unit) which utilized the same $NO_2$ sensor model, Alphasense NO2-B43F. In the laboratory testing of the study, it was observed that the amplitude of the $NO_2$ sensor response, caused by changes of relative humidity (RH), was similar to the magnitude and the rate of RH variation (Figure 2 in Mueller et al.). In addition, it was found in this study that the $NO_2$ sensor showed a delayed and exponentially decaying response upon changes in relative humidity. The term $D_{RH}$ was introduced for an approximation of this behavior and including $D_{RH}$ as a predictor variable in the calibration models largely mitigates this memory effect. In addition, the earlier study demonstrated that the model with $D_{RH}$ had lower RMSE than the model without it. In the equation, $\sum_{\Delta t=0}^{-500} \Delta S_{RH}(t+\Delta t)$ signifies the summation of RH changes in the past 500 minutes, and since the effect of RH variation exponentially decreases over time and returns to zero, the exponential term $exp(\frac{\Delta t}{\Delta t_0})$ was chosen. Various values of $\Delta t_0$ were examined during the selection of calibration models because as demonstrated in Mueller et al., RH changes in the field measurement would differ from those in the laboratory test, and the precise physical cause of this signal is unknown. The explanations in Section 2.3. have been extended and read now as follows:

In an earlier study by Mueller et al. (2017), it was observed that for the $NO_2$ sensors the amplitude of the sensor response caused by varying relative humidity is of similar magnitude than the sensor response caused by typical ambient levels of $NO_2$. In addition, it was found in Mueller et al. (2017) that the $NO_2$ sensors showed a delayed and exponentially decaying response upon changes in relative humidity. Therefore, an additional variable, $D_{RH}$, was introduced for compensation of the effect of changing relative humidity on the raw sensor signal.

$$D_{RH} = \sum_{\Delta t=0}^{-500} \Delta S_{RH}(t+\Delta t) * exp(\frac{\Delta t}{\Delta t_0}). \tag{2}$$

$\Delta S_{RH}$ represents the change in relative humidity (in %), $\Delta t$ is the corresponding time lag in minutes and $\Delta t_0$ is a time constant. Changes in relative humidity up to 500 minutes back in time are considered and weighted using the exponential term $exp(\frac{\Delta t}{\Delta t_0})$. Similar to Mueller et al. (2017), various values for $\Delta t_0$ were examined in this study (60, 90, 120, and 150 minutes) for finding the value that leads to the best performing sensor calibration models.

3. On Page 19-20, the author summarized potential reasons causing the deterioration of sensors and highlighted meteorological events and relative humidity. This paper also discusses the aging of $NO_2$ sensors but identified ozone $O_3$ as the major cause (Li et al., Characterizing the Aging of Alphasense $NO_2$ Sensors in Long-Term Field Deployments). It will be interesting to see why different reasons for sensor aging were identified.

Reply: Thanks for pointing to this interesting paper. We observed two issues during the second colocation period. As described in the paper, the sensor systems were malfunctioning during several short time periods. As described, the exact reasons for this

remain unknown. We speculate that over time the sensor housing might have lost its watertightness and humidity might have entered into the sensors or other parts such as the electronics altering temporarily the response behavior of the sensor units. Second, and independent from these malfunctioning periods, we see that the performance of the $NO_2$ sensors has significantly decreased over time. In our paper we only describe the loss in data quality without speculating on the underlying reasons. The observed degradation of the sensor performance is in agreement with the findings of Li et al. (2021). Based on Li et al. (2021), it is reasonable to argue that the observed decrease can be explained by saturation of the O3 scrubber of the $NO_2$ sensors. Annual mean concentrations of $O_3$ in Haerkingen and Zuerich (urban background) are about 21 ppb and 25 ppb, respectively, meaning that the expected lifetime of the ozone scrubber is about 13 to 17 months (comparable to the situation in Pittsburgh in the Li et al. (2021) paper). We added the following text to section 2.1:

It should be pointed out here that the used $NO_2$ sensors have an $O_3$ scrubber membrane mounted on top of the inlet to prevent the interference from ambient $O_3$. The $O_3$ scrubber has reported to have a capacity of 250 ppm h of $O_3$ (Li et al., 2021) and thus a limited lifetime.

The following text was added to section 3.2.2:
A similar degradation of the performance of the same $NO_2$ sensor has been reported by Li et al. (2021). In their study, sensor performance degradation was noticeable after 200 - 400 days of deployment, a time period that was in agreement with the expected lifetime of the $O_3$ scrubber as calculated from its reported capacity and the $O_3$ concentration at the deployment site. It is therefore reasonable to assume that the decrease in $NO_2$ sensor performance observed in this study is also influenced or caused by saturation of the $O_3$ scrubber of the $NO_2$ sensors. At the co-location site Haerkingen and in the urban background of Zuerich, annual mean concentrations of $O_3$ are about 21 ppb and 25 ppb, respectively. This means that the expected lifetime of the $O_3$ scrubber is about 13 to 17 months, which is comparable to the situation described by Li et al. (2021).

---

## Author Comment (AC2)

**Reply to the comments of Laurent Spinelle on the manuscript "Long-term behavior and stability of calibration models for NO and NO2 low cost sensors"**

**Horim Kim1, Michael Müller1,2, Stephan Henne1, and Christoph Hüglin1**

1Laboratory for Air Pollution and Environmental Technology, Empa, 8600 Dübendorf, Switzerland 2now at: Amt für Geoinformation, Kanton Basel-Landschaft, 4410 Liestal, Switzerland.

We sincerely thank the reviewer for the positive review and the constructive feedback to improve the manuscript. Please find the detailed replies to all points made by the reviewer. In the following, the reviewer's comments are given in black and the replies in blue and revised text in green.

Very interesting paper and very interesting work, particularly on this rather new subject with such a very long-term experimental data base. The work carried out is presented in a very thorough way, which in a way do not help for a fast reading but help the interested reader to fully understand the work. Congratulations to the author for this work. Only few comments and questions below:

Line 64: "two identical electrochemical" what do you mean by identical? are they coming from the same batch? or is it only that they are the same model?

**Reply:** It meant that the two NO sensors and two  $NO_2$  sensors are the same sensor model, Alphasense NO-B4 and Alphasense NO2-B43F, respectively. The term was corrected to "two electrochemical".

Line 64-65: "relative humidity sensor and a temperature sensor" is this a unique sensor? in this case maybe you can write "a combined relative humidity and temperature sensor".

**Reply:** It was a unique sensor (Sensirion STH21), which measured both temperature and relative humidity. The term was corrected to "a combined relative humidity and temperature sensor."

Line 153: "For evaluation of the sensor calibration performance", I think 1 the is missing at the beginning of the sentence "For the evaluation of the sensor calibration performance"?

**Reply:** We corrected this in the revised manuscript.**

Line 168: "An schematic", only a typo "A schematic".

**Reply: We corrected this in the revised manuscript.**

Paragraph 3.1.2: did you considered to filter based on the manufacturer's limit of detection or one you could have evaluated with some lab test ? in fact, it is known that at low ambient air concentration (10 - 15 ppb) sensors response is dominated by noise or interference.

**Reply:** The technical specification provided by Alphasense quantifies the sensor noise of  $\pm 2$  standard deviations as 15 ppb for both the NO2-B43F and NO-B4 sensor, when testing the sensors in 'Alphasense ISB low noise circuit'. However, the criteria might not be applicable to the data collected from the field measurement because the sensor response in the ambient environment may not be comparable with the response during laboratory tests. In the current study, we did not filter any raw data except the sensor malfunction periods occurring during the second co-location period, to show the full range of the sensor measurement in the field condition.

no changes in the revised text.

Line 304-305: "the penetration of raindrops into the sensor units may cause significant disturbance of the sensor signal", do you mean disturbance on the electronic components?

**Reply:** Yes. It meant that the raindrops may penetrate into the housing of the sensor units during rain events and hinder the signal transfer in the electronic component of the sensor. The sentence was corrected to "the penetration of raindrops into the housing of the sensor units may cause significant disturbance of the sensors or other components of the sensor units."

Line 346: "lower, medium and higher  $NO_2$  levels", you should maybe give your range of concentration as those categories may vary a lot from country to country.

**Reply:** The sentence intended to point out that the  $NO_2$  sensor performance during their deployment in a small sensor network in the city of Zurich allowed distinguishing sites with lower (AC009 and AC010), medium (AC011), and higher (AC012)  $NO_2$  levels as shown in Figure 7. According to the measured ranges from the sensor units, the ranges are now stated in the sentence as "with lower (0 - 20 ppb), medium (20 - 30 ppb) and higher (40 - 50 ppb)  $NO_2$  levels as shown in Figure 7."

End of the conclusion: Do you consider the contrast you are pointing out between NO and  $NO_2$  can be inked to the difference between the gaseous species involved and there sensitivity to interferent ? e.g.  $O_3$  is a well-known interferent for  $NO_2$  sensors which can impact drastically the data quality, in particular in the filtered sensor version for which the filter efficacity depend on the  $O_3$  level, whereas this kind of strong interference are less common for NO sensors.

**Reply:** Thanks, this is a very valid point that has also been addressed by referee #1. As described in our reply to the third comment of referee #1, we now mention the interference to  $O_3$  as a reasonable and likely explanation for the observed ageing of the NO2 sensors and the absence of a clear degradation over time of the NO sensors. In the revised manuscript we refer now to the very relevant paper by Li et al. 2021 as mentioned by referee #1. See our reply to the third comment of referee #1 for details and the changes made in the revised manuscript.

---

## Author Response (AR2)

**AMT-2021-433 Author's response to Associate Editor's comment on the revised manuscript.**

The authors would like to thank the associate editor for your suggestions for manuscript correction. Below we demonstrated the addressed issues and the relevant changes.

**1. The revised manuscript seems to address all of the reviewer's concerns. It would be good to clarify one item about the target plots. Since both CRMSE (as defined in Table 2) and sigma_y are positive numbers, it seems like the x-axis cannot span the range -1:1. However, it seems like the convention with these plots is to plot the CRMSE/sigma as <0 when the standard deviation of the reference data are larger than the standard deviation of the calibration model output (e.g., see Zimmerman et al Figure 7). It would be good to specify this convention in your manuscript.**

A. The convention on the sign of CRMSE/sigma has already been mentioned in the manuscript line 172-173 and line 176-177, Section 2.4.3. To specify this convention in figures with target plots, We added the following text to the caption of Figure 5 in newly revised manuscript:

"Note that by convention the sign of CRMSE/$\sigma_y$ in target diagrams is determined by the sign of ($\sigma_{\hat{y}} - \sigma_y$)."